# Tissue Characterization in Cardiac Amyloidosis

**DOI:** 10.3390/biomedicines10123054

**Published:** 2022-11-28

**Authors:** Veronica Musetti, Francesco Greco, Vincenzo Castiglione, Alberto Aimo, Cataldo Palmieri, Dario Genovesi, Assuero Giorgetti, Michele Emdin, Giuseppe Vergaro, Liam A. McDonnell, Angela Pucci

**Affiliations:** 1Health Science Interdisciplinary Center, Scuola Superiore di Studi Universitari, S.Anna, 56127 Pisa, Italy; 2Fondazione Toscana G. Monasterio, 56124 Pisa, Italy; 3Fondazione Pisana per le Scienze, 56017 Pisa, Italy; 4Histopathology Department, Pisa University Hospital, 56124 Pisa, Italy

**Keywords:** cardiac amyloidosis, transthyretin amyloidosis, AL amyloidosis, histology, immunohistochemistry, mass spectrometry, endomyocardial biopsy, abdominal fat tissue biopsy

## Abstract

Cardiac amyloidosis (CA) has long been considered a rare disease, but recent advancements in diagnostic tools have led to a reconsideration of the epidemiology of CA. Amyloid light-chain (AL) and transthyretin (ATTR) amyloidoses are the most common forms of cardiac amyloidosis. Due to the distinct treatments and the different prognoses, amyloid typing is crucial. Although a non-biopsy diagnosis can be obtained in ATTR amyloidosis when certain diagnostic criteria are fulfilled, tissue characterization still represents the gold standard for the diagnosis and typing of CA, particularly in AL amyloidosis. The present review focuses on the status of tissue characterization in cardiac amyloidosis, from histochemistry to immunohistochemistry and mass spectrometry, as well as on its future directions.

## 1. Introduction

Amyloidoses are characterized by the extra-cellular deposition of amyloidogenic proteins in a cross-β-sheet structure that confers a typical green birefringence by Congo red staining under polarized light microscopy [1]. The tissue deposition of amyloid may lead to organ failure [1,2]. Cardiac amyloidosis (CA) has long been considered a rare disease [1,3]; however, recent advances in non-invasive diagnostic tools have led to a reconsideration of the epidemiology of the disease. Furthermore, the new therapeutic strategies available have increased the clinical impact of CA diagnosis [4,5,6,7,8].

More than 30 amyloidogenic proteins have been identified so far, with immunoglobulin light chains and transthyretin (TTR) being the most common amyloidogenic precursors of CA, respectively reasonable for AL and ATTR amyloidosis [9,10]. AL amyloidosis is a systemic disease involving almost every organ and has a poor prognosis, particularly when the heart is affected [11]. Two major forms of ATTR amyloidosis are recognized: wild-type ATTR (ATTRwt) and variant ATTR (ATTRv; due to point mutations in the *TTR* gene) amyloidosis [12,13]. In CA, the myocardial damage and the functional impairment are related to the type of the amyloidogenic protein as well as to the pattern and the extension of the amyloid deposits [11]. Recent investigations have also identified other mechanisms, including fibrosis and inflammatory pathways, which may play a significant role in myocardial dysfunction [14,15].

Amyloid typing is crucial for the clinical management of affected patients, since the misidentification of the amyloidogenic protein can lead to inappropriate treatments [7,16]. In some cases, particularly in ATTR amyloidosis, a non-biopsy diagnosis can be obtained; nevertheless, tissue characterization is always needed when a plasma cell dyscrasia is present and/or when nuclear medicine findings are inconclusive [17,18,19].

The present review provides an overview of the status of tissue characterization in CA and its future directions, considering its impact on the clinical management of patients.

## 2. Tissue Biopsy for the Diagnosis of Amyloidosis

Tissue biopsy represents the gold standard for the diagnosis and typing of amyloid deposits [7,20]. The development of new imaging techniques has dramatically contributed to the diagnosis of CA and non-biopsy diagnosis can be achieved in ATTR CA through the demonstration of myocardial uptake by ^99m^Tc-hydroxymethylene diphosphonate (^99m^Tc-HMDP) scintigraphy together with echocardiographic or CMR features consistent with amyloidosis in the absence of monoclonal gammopathy [17,21,22,23] (Figure 1).

The biopsy of a clinically involved organ represents the most sensitive method to diagnose amyloidosis and makes it possible to investigate possible concomitant pathologies [15,24]. In the localized forms of amyloidosis, tissue biopsy is performed on the affected organ or tissue, whereas in systemic amyloidosis, other sites such as abdominal fat, the gastro-intestinal tract including the rectum, bone marrow or minor salivary glands may be biopsied, with the positive diagnostic yield depending on the involvement of the surrogate site and on the type of amyloid deposits [7]. The biopsy of a surrogate site is often recommended for the diagnosis of cardiac amyloidosis because organ biopsy could be risky for the patient and require a technical expertise [24]. However, a negative result at a surrogate site does not exclude the diagnosis and if the clinical suspicion is high, the biopsy of the involved organ should be performed [25].

### 2.1. Abdominal Fat Biopsy

Periumbilical fat biopsy is widely used for the diagnosis of systemic amyloidosis [26,27]. A periumbilical fat biopsy with evidence of amyloid deposits together with imaging features consistent with heart involvement makes it possible to diagnose CA [28]. Abdominal fat fine-needle biopsy [29] and abdominal fat pad excisional biopsy (FPEB) [30] have been used for the diagnosis and characterization of cardiac amyloidosis. Fat biopsy was first introduced by Westermark and Stenkvist in 1973; they described abnormal fat aspirates in 9 out of 28 patients with suspected systemic AA amyloidosis [31]. The main advantages of this technique are its simplicity, low cost, high patient tolerance and lack of significant complications [7]. In patients with suspected cardiac amyloidosis, abdominal fat fine-needle aspiration biopsy has elevated specificity (100%) and sensitivity for AL CA, with positive results for amyloid in up to 84% of cases (Figure 2), whereas it is far less sensitive for ATTRv (positivity in <45% of patients) and for ATTRwt (about 15% of cases) amyloidoses [27,32]. Nevertheless, it must be noted that fat biopsy may miss the diagnosis even in AL cases, a negative result not excluding AL diagnosis [33]. The amyloid deposits are usually thin and interspersed within the fibro-fatty tissue that is obtained by periumbilical abdominal fat aspirate, with peri-cellular, septal or vascular localization, but they may also exhibit a nodular pattern [30].

However, an important issue to underline is specimen adequacy. Guy et Al. evidenced a significant rate (11%) of inadequate samples in fine-needle aspirates and suggested the visual inspection of fat fragments (to exclude possible oil droplets) at the time of aspirate collection [33]. Tissue fragments also require careful sampling after fixation because fatty fragments tend to float in the fixative and may easily be lost during processing. The refrigeration of the samples before their removal from formalin has also been suggested to avoid possible loss of tissue [35]. Another limit of this technique is the tissue size, which could provide false negative results or be inadequate for amyloid typing [36]. FPEB is only slightly more invasive than fine-needle aspiration biopsy, but it can evidently provide more adequate material for amyloid typing [26]. The sensitivity of FPEB for AL amyloidosis depends on biopsy size, ranging from 50% for biopsies less than 700 mm^3^ to 100% for biopsies greater than 700 mm^3^ [30]. Due to the fact that the fat biopsy can sometimes be split up for different investigations, including immunofluorescence, mass spectrometry and electron microscopy, it could be advisable for the initial surgical biopsy to have a dimension of at least 1400 mm^3^ [30]. So far, no study has fully investigated and clarified how much FPBE size could influence the diagnostic sensitivity.

### 2.2. Endomyocardial Biopsy

Endomyocardial biopsy (EMB) can be performed in patients with suspected CA when a peripheral biopsy is inconclusive [20,26,37]. EMB is performed in experienced centers, and it is mostly performed in the right ventricle (RV). Nevertheless, left ventricular (LV) EMB may be more sensitive to detect the earlier sub-endocardial histological changes as compared to RV EMB [38]. In qualified centers, LV EMB presents a low risk of complications, particularly when performed with a trans-radial approach [38,39]. On EMB, the amyloid deposits are mostly detected in the interstitial space, surrounding the myocytes (see Section 3) [38].

### 2.3. Other Biopsy Sites

Other tissues may undergo biopsy and provide adequate samples for the diagnosis and characterization of amyloid deposits, particularly in systemic forms such as AL amyloidosis [8,19,26]. The diagnostic yield is related to the extension of the disease.

Bone marrow biopsy is part of the routine evaluation of patients with suspected AL amyloidosis, amyloid deposition being detected in 50–60% of such cases [7,40].

Minor salivary gland biopsy, i.e., the removal of one or more minor salivary glands from the labial mucosa, has also been used for the diagnosis of AL amyloidosis, with a sensitivity up to 86% [41].

In gastrointestinal biopsies, amyloid is mostly found in elderly male patients with the AL type [42]. The study from the “Amyloid Registry Kiel” showed that different types of amyloids could have distinctive deposition patterns; biopsies of the upper gastrointestinal tract were more often positive in AL κ+ forms and in AA amyloidosis, whereas biopsies from the large intestine and rectum (enclosing the submucosal layers) were more frequently positive in AL λ+ forms and in ATTR [42]. The rectal biopsies may show amyloid deposits in the *muscularis mucosae* and submucosa in both AL (86% sensitivity) and ATTR amyloidosis (the positivity being limited to ATTR amyloidosis patients with gastrointestinal symptoms), but they have a quite limited significance for the initial diagnosis of amyloidosis because they are often negative in patients with a negative abdominal fat biopsy [43,44].

Other sites, such as the gingiva or skin, are far less sensitive and rarely used in the clinical practice for amyloid typing [7].

## 3. Histology of Cardiac Amyloidosis

At histology, large amyloid deposits are shown as eosinophilic and amorphous aggregates on hematoxylin and eosin (HE) stained histological sections, whereas subtle and initial amyloid deposits, either interstitial or within intra-myocardial vessel walls, are difficult to detect (Figure 3).

Amyloid fibril deposits are mainly interstitial and surround the myocytes, which may be atrophic and undergo progressive loss. Nodular deposition can also be found [11,15]. Morphology does not make it possible to identify amyloids and to accurately differentiate the different forms of amyloidosis subtypes [45,46]. However, a few morphological differences have been described. In AL amyloidosis, the amyloid deposits are mainly pericellular and reticular, often extending to >40% of myocardial tissue; they often show the presence of inflammatory infiltrates, mainly T-lymphocytes and macrophages, which may contribute to tissue damage [11,14,15,47]. Conversely, in ATTR amyloidosis, amyloid deposits are usually irregular and patchy, displaying two main patterns, i.e., nodular deposits together with diffuse interstitial deposition (pattern A), or thin interstitial and vascular deposits (pattern B) [48]. The fibrils are misaligned and rather short in pattern A, containing the C terminal fragments of TTR, while in pattern *B* they are longer, within thinner and well-delimited bundles [48]. So far, type B fibrils (full-length TTR fibrils only) have been observed in ATTRv, whereas type A fibrils (a mixture of truncated and full-length TTR) may be present in either ATTRv or ATTRwt [9,49].

## 4. Histochemistry of Cardiac Amyloidosis

On light microscopy, amyloid deposits can be suspected due to their morphology (Figure 4A, asterisks), but they must be demonstrated by a specific histochemical staining for amyloid; the most used Congo red staining highlights the amyloid as a red or salmon-pink substance that shows a characteristic apple-green birefringence under polarized light (Figure 4B) [50].

The tissue must be examined by an experienced pathologist, confidently knowledgeable about the disease, the histological techniques and their possible pitfalls. Endomyocardial biopsy is an invasive technique that must be performed in referral centers, adequately processed by using standardized protocols (including serial sections of biopsy fragments) and examined by a pathologist specialized in cardiovascular pathology [20]. The number of biopsy fragments is crucial for avoiding false-negative results (at least three formalin-fixed and paraffin-embedded samples, undergoing serial sections) due to the possible focal distribution of amyloid deposits, particularly in initial diseases [15,20]. Congo red staining, i.e., the most widely used histochemical technique for evidencing amyloid deposits, must be performed on 8–9 µm thick sections to increase sensitivity and must be examined under polarized light to differentiate the collagen, fibrin or red blood cells white birefringence from the apple-green birefringence of amyloid fibrils. It is also advisable to have a positive control for amyloid on the same Congo red-stained histological slide of the testing tissue biopsy. Masson’s trichrome staining highlights the connective fibrous tissue with intense blue staining. It may be quite helpful (albeit not necessary) to discriminate amyloid vs. fibrosis extent, as it gives a blue-violet appearance to amyloid deposits (Figure 4C).

The distribution of Congo red positivity is quite variable, depending on the extent of the amyloid deposits, which may be interstitial and/or nodular, focal or diffuse, peri-vascular and sometimes into the vessel wall (Figure 5).

## 5. Immunohistochemistry of Cardiac Amyloidosis

Immunohistochemistry is commonly used for amyloid typing, but it requires dedicated expertise, availability of specific antibodies and a rigorous standardization of the protocols; otherwise, it may provide inconclusive or even misleading results [7,51,52]. First, the specific antisera/antibodies against the different possible types of amyloids must be tested on the myocardial tissue to avoid false-negative results, mainly depending on the characteristics of the specific antisera/antibodies or inappropriate experimental conditions. Tissue processing, effective antigenic unmasking, as well as the quality of the available antibodies, are essential to obtain optimal immunostaining and a high rate of reliable results, particularly on formalin-fixed and paraffin-embedded (FFPE) biopsies [52]. Over-fixation may interfere with tissue antigenicity and give misleading results, mostly causing a high background that may be erroneously interpreted as a false positive. It is advisable to avoid the over-fixation of biopsies by processing them on the same day, after a few hours of formalin fixation. The use of well-tested antibodies and an automated immunostainer with standardized and reproducible protocols may also help to sensibly improve the results and their reproducibility [15,51,52,53,54]. Immunoperoxidase is the most used technique in clinical practice, albeit immunofluorescence can also be applied, particularly if the samples are intended to be analyzed with confocal microscopy [53].

Lambda light chains represent the most common amyloidogenic precursors in AL amyloidosis. In such cases, positive immunostaining only for the lambda light chain is detected, i.e., the so-called monotypic restriction [10,15,18,28,29,54,55] (Figure 6 and Figure 7A).

In ATTR CA, the immunostaining for TTR must be intense and consistent with Congo red-positive amyloid deposits (Figure 7B), without monotypic restriction for immunoglobulin light chains.

“Dual pathology” cases represent an extremely rare exception, as shown in Figure 8 which illustrates the co-existence of ATTR and immunoglobulin light chain Lambda immunoreactivity in an endomyocardial biopsy, the immunohistochemical findings being confirmed by imaging that show myocardial uptake by diphosphonate (DPD) scintigraphy and positive ^18^F-florbetan positron emission tomography (PET) [15,56].

## 6. Mass Spectrometry-Based Proteomic Analysis of Cardiac Amyloidosis

In recent years, the mass spectrometry-based proteomic analysis of amyloid deposits has been established for amyloid subtyping [58,59]. Amyloid typing using mass spectrometry-based proteomics was first reported in 2008 [60,61]. The direct identification of the proteins in the amyloid deposits is unbiased [60], overcomes the need for a specific antibody for each potential amyloidogenic protein and is largely insensitive to the 3D structure of the aggregated proteins (which affects antibody–protein binding) [62]. The first demonstrations of amyloid typing with mass spectrometry-based proteomics used the ionization technique matrix-assisted laser desorption/ionization (MALDI) to identify the proteins previously separated by a 2D gel [61]. Today, the vast majority of mass spectrometry-based proteomics experiments are performed using liquid chromatography coupled with tandem mass spectrometry (LC–MS/MS) on account of its greater speed and sensitivity [62].

The most common samples that are used for the typing of cardiac amyloidosis are the subcutaneous fat aspirates and the EMBs [63]. In these latter ones, amyloid deposits are usually better defined than in fat tissue where they are more dispersed within the extracellular space between adipocytes and more difficult to isolate. Subcutaneous fat sampling is far less invasive than a cardiac biopsy EMB, but its diagnostic potential, particularly for ATTRwt, is more limited due to the lower number of amyloid deposits [27,64]. For both types of specimens, i.e., EMB and subcutaneous fat, washing the samples is crucial since blood residues can introduce contamination due to the presence of soluble amyloidogenic precursors in the bloodstream [59,65]. Moreover, sample heterogeneity should always be taken into consideration; therefore, the presence of amyloid deposits should be verified before molecular analysis, for instance by using Congo red staining.

In most cases, biopsies are FFPE, although they may be processed without any chemical fixation. FFPE specimens have the significant advantage that they can be stored at room temperature and produce higher quality histological sections. For these reasons, FFPE samples are, by far, the most common format in clinical practice and in clinical biobanks. Formalin fixation preserves the sample by producing methylene bridges (or cross-links) between the amine groups present in proteins [66]. Thus, chemical modifications due to methylation should be considered when searching for peptides from amyloidogenic proteins in FFPE samples [67]. The network of cross-links between the proteins in FFPE samples makes protein extraction more laborious and time consuming. A non-prolonged formalin fixation (few hours) is then recommended for samples undergoing proteomics analysis [67,68,69].

Currently, there is no standard operating protocol for amyloid typing since the sample processing, the instrumental setup and even the data analysis pipeline depend on the specimen type and the instrumentation availability [70].

The sample processing protocol for proteomics-based diagnosis depends on the type of specimen. While some protocols are available for formalin fixation and embedding of subcutaneous fat aspirates [71], this tissue is not particularly suitable to produce FFPE tissue blocks due to its lack of compactness. Subcutaneous fat is usually processed directly, via tissue solubilization and digestion [65,72,73,74]. Cardiac biopsies are usually fixed and embedded, and multiple tissue sections can be obtained from a single block. In order to focus the LC–MS/MS analysis on the amyloid deposits, and thereby increase specificity and sensitivity, the isolation of amyloid deposits by laser-capture microdissection (LCM) was introduced [58,60]. Tissue sections are mounted on membrane-coated glass slides and stained by Congo red. Congo red-positive areas, confirmed by Congo red autofluorescence, are selected and precisely cut by the instrument. The selected areas are transferred into the cap of a tube for subsequent proteomics sample preparation and analysis [58,60]. An area of 50,000 μm^2^, isolated from a 8–9 μm thick section, is sufficient for complete amyloid characterization [58,60]. LCM increases the sensitivity of the analysis by reducing the contribution of proteins from the cellular background of the tissue. Protein extraction from FFPE samples involves prolonged heating in the presence of surfactants (90 min to overnight) [58,60,74,75,76], while adipose tissue is usually sonicated for a shorter period of time (15 s to 30 min) [72,73]. Dialysis [66] or molecular filters [74] can be used to remove surfactants used for tissue solubilization and to exchange the buffer to one better suited for proteolytic digestion. Disulfide bonds can be reduced and alkylated [74,75] to aid protein denaturation and thereby increase the accessibility of the protease to the protein backbone. Proteins are then digested using a protease, typically trypsin, at 37 °C overnight [75,76] or, for shorter turn-around times, for 2 h [65,74]. Interestingly, some protocols do not include protein reduction and alkylation [65] or perform the reduction after the tryptic digestion [74]. The peptides produced by proteolytic digestion, often referred to as tryptic peptides, are then purified using C18 solid phase extraction columns, vacuum-dried and resuspended for the chromatographic injection [74,75]. A summary of the workflow is shown in Figure 9.

Tryptic peptides are analyzed by liquid chromatography coupled with mass spectrometry. The most common setup consists of a C18 chromatography coupled with the mass spectrometer by an electrospray ionization source [74,75,76], but two-dimensional chromatography has also been used to increase the number of detected peptides [56,59,66]. Most mass spectrometry methods used for amyloidosis typing are data-dependent analyses (DDA) [58,60,65,73,75,76]. In a DDA experiment, the masses and intensity of the peptides (eluting from the LC column) are measured in an MS1 spectrum; the most intense peptide ions are automatically selected for isolation and fragmentation. The masses of the peptide fragments are then measured, the so-called MS2 or MS/MS spectrum. Since peptides fragment in a predictable manner, e.g., b- and y- fragment ions for collision-induced dissociation [77], the combination of fragment ion masses and the mass of the precursor ion can be used to identify the peptide.

DDA does not require the prior specification of target peptides masses, and thus allows for the analysis of the whole proteome content of the sample. However, DDA preferentially selects peptide ions of higher intensity and is therefore biased toward more abundant peptides. The stochastic nature of peptide ionization and peptide selection based on intensity in DDA can become important when attempting to identify lower abundant peptides; in short, lower abundant peptides may not be selected for fragmentation and thus will not be identified. The high scanning speed of modern LC–MS/MS instruments (>50 peptides can be selected and fragmented each second) has reduced the severity of this effect and increased the number of proteins that can be identified in a single LC–MS/MS experiment.

Conti et al. proposed a targeted approach to distinguish between AL-κ, AL-λ and ATTR by monitoring the fragments of the characteristic peptides for the three amyloidogenic proteins [74]. Targeted approaches can be performed on less expensive instruments and are simpler to standardize. Nonetheless, targeted approaches are blind to peptides that are not included in the target list, and are thus not effective in detecting new amyloidogenic proteins nor unexpected mutant forms of the proteins.

Peptide identification in DDA analysis is based on a comparison of the fragment ions detected in the LC–MS/MS experiments with those expected to be produced by reference peptides. A database of reference proteins is digested in silico and a list of reference tryptic peptides is generated. MS2 spectra are matched against the expected fragmentation spectra of the reference peptides [77] and the identified peptides are grouped into proteins. Relative protein quantification is performed by summing the MS1 intensities of the peptides of each protein. For this reason, a larger number of MS2 spectra allows more peptides to be identified, permitting the investigation of lower-abundance proteins. The exploration of low-abundance proteins may be important in the context of early diagnosis or to detect small deposits in subendothelial fat samples [70].

There is no standardized method to generate a diagnosis from the list of quantified proteins. A common method is to identify the amyloidogenic protein detected with the largest number of MS2 spectra [58,60]. This approach is simple and straightforward, especially for LCM-isolated deposits. In the presence of blood contamination, circulating amyloidogenic proteins can mask the proteins from the amyloid deposit. This problem is particularly pertinent for subendothelial fat samples because the dispersed nature of the amyloid deposits means the amyloid proteins from the deposits are less abundant. A method to reduce the deleterious effect of blood contamination on amyloid typing has been reported, in which a comparison of the proteomics profile of the test sample with healthy controls was used to reduce the contribution of circulating amyloidogenic proteins to the diagnosis [65]. Apolipoprotein E, apolipoprotein A-IV and serum amyloid P-component co-precipitate with the amyloidogenic protein and are thus considered a signature for amyloid deposit [73,74]. The presence of these proteins was investigated as a universal protein signature of amyloidosis and exhibited a better performance than Congo red in the diagnosis of amyloidosis from subcutaneous fat [73].

## 7. Immuno-Electron Microscopy of Cardiac Amyloidosis

Transmission electron microscopy (TEM) and post-embedding immunogold techniques are currently used in specialized centers and may prove very useful for the detection and typing of amyloid deposits [29,78]. TEM is particularly useful in the early detection of amyloid due to its high resolution, although the focal and irregular distribution of amyloid deposits may limit its sensitivity.

For TEM, a very small biopsy sample is sufficient, which attributes advantage and value to this technique. The tissue sample must be fixed by a 2.5% glutaraldehyde or 2% Karnowsky (i.e., 2% paraformaldehyde and 2% glutaraldehyde) solution and embedded in Epon-Araldite resin, after eventual post-fixation in osmium tetroxide. On ultrathin sections, the amyloid fibrils can be morphologically identified after routine 5% uranyl acetate and Reynold’s lead citrate staining, whereas amyloid typing can be performed by using the post-embedding immunogold technique [29]. Briefly, ultrathin sections are incubated with specific primary antibodies (against TTR, kappa or lambda light chains) and, after appropriate washing, the reaction is detected by the subsequent incubation with an appropriate secondary, anti-mouse (GAM) or anti-rabbit (GAR) antibody bound to colloidal gold particles of 5 to 15 nm diameter (Figure 10).

## 8. Immunofluorescence and Confocal Laser Scanning Microscopy

The fluorescence of Congo red staining and immunofluorescence staining (using different fluorochromes and specific antisera/antibodies raised against amyloidogenic proteins) can be detected and analyzed by confocal microscopy [53]. Confocal laser scanning microscopy (CLSM) makes it possible to obtain a tri-dimensional reconstruction of EMB samples with a submicrometric resolution, so it may help better describe the tissue architecture, the possible co-localization and the spatial interactions of amyloidogenic precursors and other molecules in the myocardium. CLSM can also improve the sensitivity of Congo red staining to detect initial or limited amyloid deposits and to correctly localize them within the tissue [79].

## 9. Future Directions

The need for early diagnosis of CA and accurate amyloid typing is raising new problems and challenges for both physicians and pathologists [80,81,82]. The presence of a dual pathology, i.e., ATTR and AL amyloidosis developing subsequently in the same patient and affecting the heart, has been demonstrated in anecdotic cases, but the advancement in diagnostic tools will likely disclose more of these forms, which will then raise relevant issues for treatment strategies [15,56,83].

Histochemistry and immunohistochemistry are the most common and widely used methods for diagnosing and typing CA, their sensitivity and reliability being quite high in experienced centers [15,20,29,52], but mass spectrometry-based proteomics has recently become the gold standard technique for amyloidosis typing due to its high diagnostic accuracy [63,84]. Both histochemistry and immunohistochemistry need pathology expertise and must be performed according to standardized protocols to avoid pitfalls, misdiagnosis or inconclusive results, first by the proper fixation of tissue samples and possibly automated staining (which are more easily standardized and reproducible) with appropriate controls [15,52]. Red Congo-guided LC–MS analyses can be performed on FFPE biopsies (mainly EMB), allowing not only a sensitive diagnosis of CA, but also the characterization of amyloid deposition and the comparison with the histological features. The availability of such techniques may help in the study of etiopathogenetic mechanisms in CA; recent studies have shown that fibrosis and inflammatory infiltrates may be associated with amyloid deposits in CA and may influence cardiac dysfunction [14,15]. The proteomics assessment of amyloid deposit composition may also detect new amyloidogenic proteins and pathogenic mutations [84,85]. A standardization of the proteomic protocols (that is still lacking) would ease the comparison of the results across different laboratories and a more diffuse adoption of proteomics-based amyloid typing. The implementation of new microproteomics pipelines to increase sensitivity may aid earlier amyloidosis diagnosis and treatment, such as the analysis of subcutaneous fat, which is currently rather difficult and sometimes inconclusive by the currently available proteomic analyses [71].

## 10. Conclusions

Tissue characterization is often crucial for the identification of CA. It may help to disclose and investigate peculiar forms, such as dual pathology, or initial disease, but it is also necessary to provide new diagnostic tools as well as new clues to disclose the unknown pathogenetic mechanisms of this complex and still little-known disease.

## Figures and Tables

**Figure 1 biomedicines-10-03054-f001:**
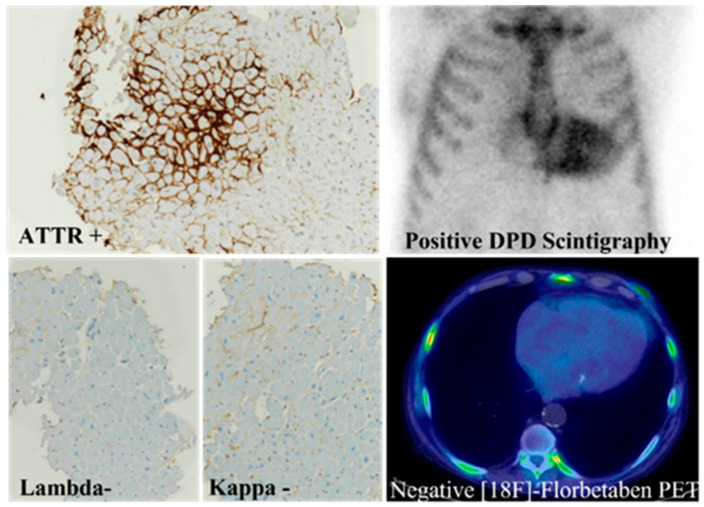
Histological evidence of ATTR amyloidosis. ATTR immunoreactivity (ATTR+) in the endomyocardial biopsy of a patient showing positive diphosphonate (DPD) scintigraphy, together with negative immunostaining for immunoglobulin light chains (Lambda-, Kappa-) and negative ^18^F-florbetaben positron emission tomography. Immunoperoxidase with hematoxylin counterstaining (ATTR+, Lambda-, Kappa-); original magnification 4× (modified from Figure 1i of ref. [15]).

**Figure 2 biomedicines-10-03054-f002:**
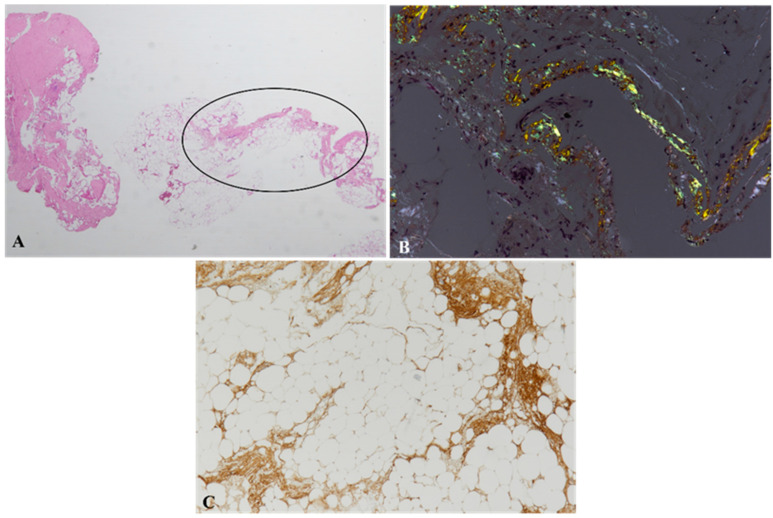
Abdominal fat biopsy. In a patient with AL Lambda+ cardiac amyloidosis, a limited area (evidenced by the circle, (**A**)) of the biopsy shows apple green birefringence on Congo red staining under polarized light (**B**). Immunohistochemistry reveals immunoglobulin light chain lambda positivity (**C**). Original magnification: 2× ((**A**), hematoxylin and eosin) and 10× ((**B**), Congo red staining; (**C**), immunoperoxidase technique with hematoxylin counterstaining) [34].

**Figure 3 biomedicines-10-03054-f003:**
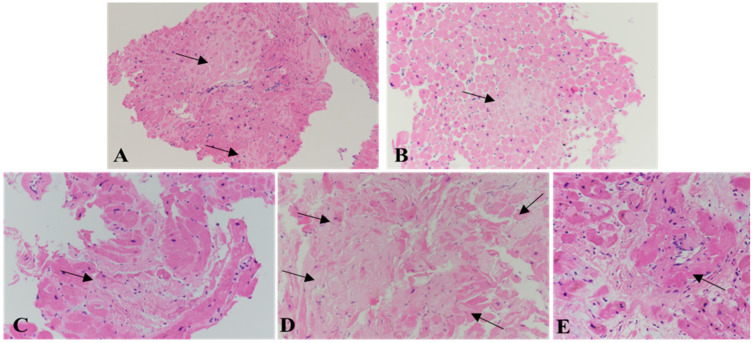
Morphology of cardiac amyloidosis. The amyloid deposits may be very focal in the initial forms of cardiac amyloidosis ((**A**,**B**); arrows) or much more diffuse, mainly interstitial and less commonly nodular ((**C**,**D**); arrows) or in the wall of intra-myocardial vessels ((**E**); arrow), with progressive loss of myocytes. Hematoxylin and eosin staining. Original magnification, 4× (**A**–**D**), 10× (**E**) [34].

**Figure 4 biomedicines-10-03054-f004:**
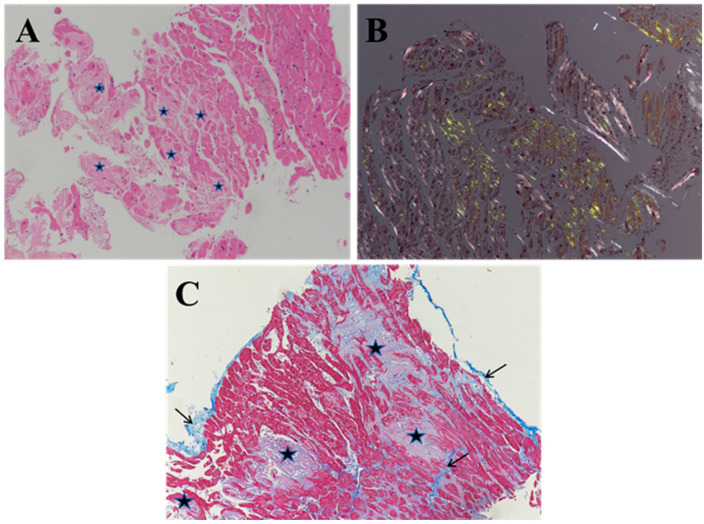
Histology and histochemistry of cardiac amyloidosis on endomyocardial biopsy. (**A**) Amorphous and eosinophilic deposits (asterisks) are evidenced in the endomyocardial interstitial spaces by histology on hematoxylin and eosin-stained slides. (**B**) They are constituted by amyloid deposits with green birefringence (as compared to the white refringence of fibrosis) on the Congo red staining (performed on an adjacent serial section) under polarized light. (**C**) By using Masson’s trichrome staining, the connective fibrous tissue (arrows) is evidenced by an intense blue staining, whereas amyloid deposits show a blue-violet color (asterisks). Original magnification, 10× (**A**–**C**) [34].

**Figure 5 biomedicines-10-03054-f005:**
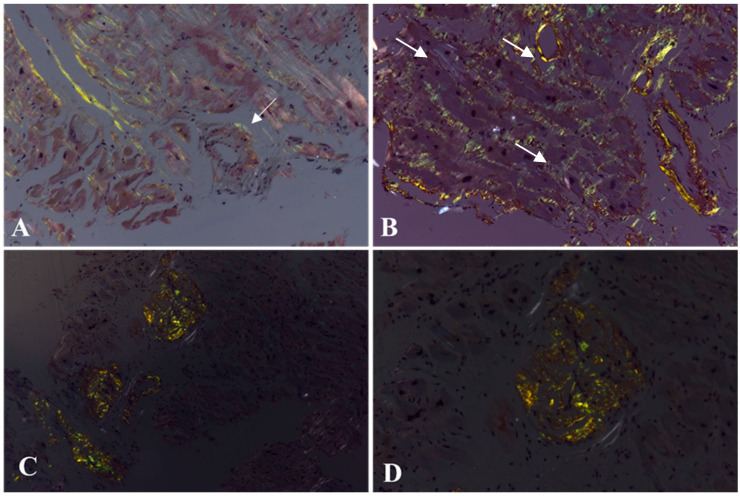
Pattern of Congo red staining positivity. The amyloid deposits show green birefringence under polarized light that may be quite focal (**A**) or plurifocal/diffuse (**B**–**D**), with an interstitial (**A**,**B**) or nodular (**C**,**D**) pattern. It may also be localized at the perivascular level ((**A**), arrow) or in the intramyocardial vessel wall ((**B**); arrows). Congo red staining under polarized light. Original magnification, 10× (**A**–**D**) [34].

**Figure 6 biomedicines-10-03054-f006:**
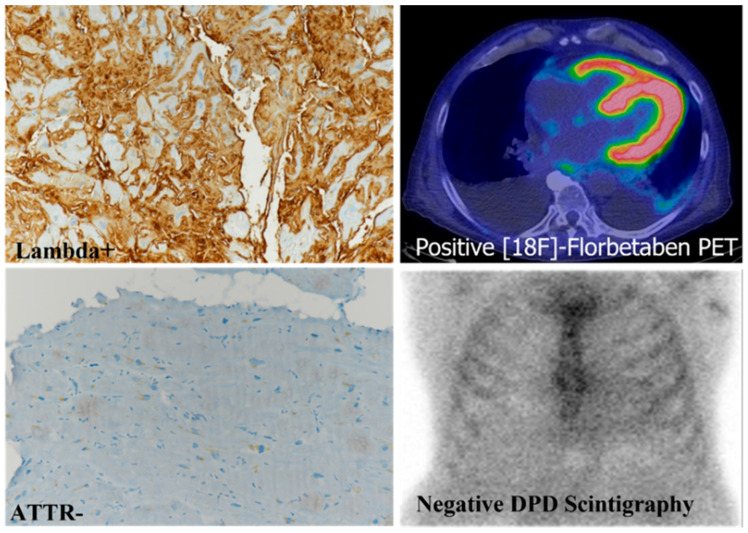
Immunohistochemistry and imaging in a Lambda-positive AL cardiac amyloidosis. Immunohistochemistry showing immunoglobulin light chain Lambda immunoreactivity with negative ATTR (ATTR-) immunostaining; the patient had a positive ^18^F-florbetaben positron emission tomography (PET) and a negative diphosphonate (DPD) scintigraphy. Immunoperoxidase staining and hematoxylin counterstaining (Lambda+, TTR-); original magnification, 10× (modified from Figure 1n of ref. [15]).

**Figure 7 biomedicines-10-03054-f007:**
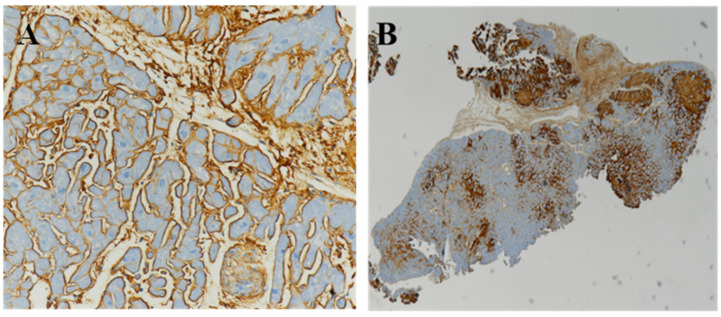
Immunohistochemistry of cardiac amyloidosis on endomyocardial biopsies. (**A**) In a patient with amyloid light chain (AL) amyloidosis, immunohistochemistry shows intense immunostaining for immunoglobulin Lambda light chain by immunoperoxidase technique. (**B**) In a patient affected by amyloid transthyretin (ATTR) amyloidosis, immunohistochemistry shows intense, interstitial and nodular immunoreactivity for transthyretin. Immunoperoxidase staining and hematoxylin counterstaining. Original magnification, 10× (**A**) and 4× (**B**) [34].

**Figure 8 biomedicines-10-03054-f008:**
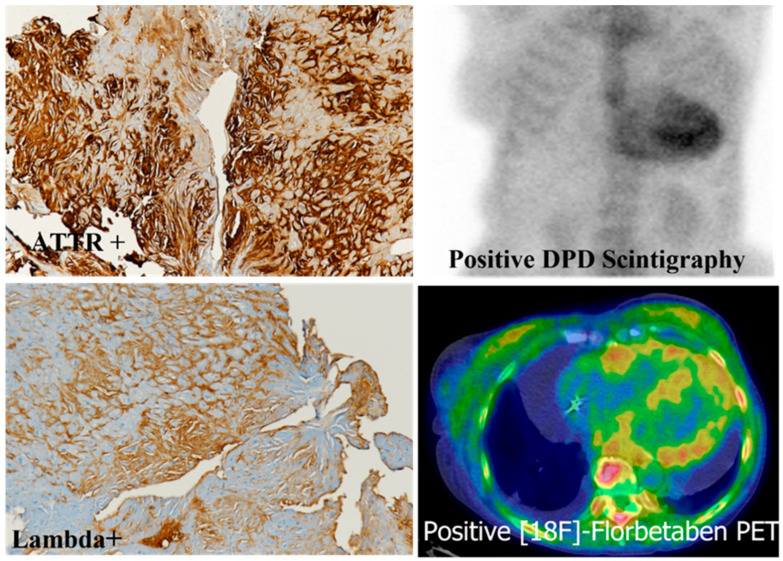
Dual pathology. Immunoreactivity for both transthyretin and immunoglobulin light chain Lambda in a patient showing myocardial uptake by diphosphonate (DPD) scintigraphy and positive ^18^F-florbetan positron emission tomography (PET). Immunoperoxidase staining with hematoxylin counterstaining (ATTR+, Lambda+), original magnification 10× [57].

**Figure 9 biomedicines-10-03054-f009:**
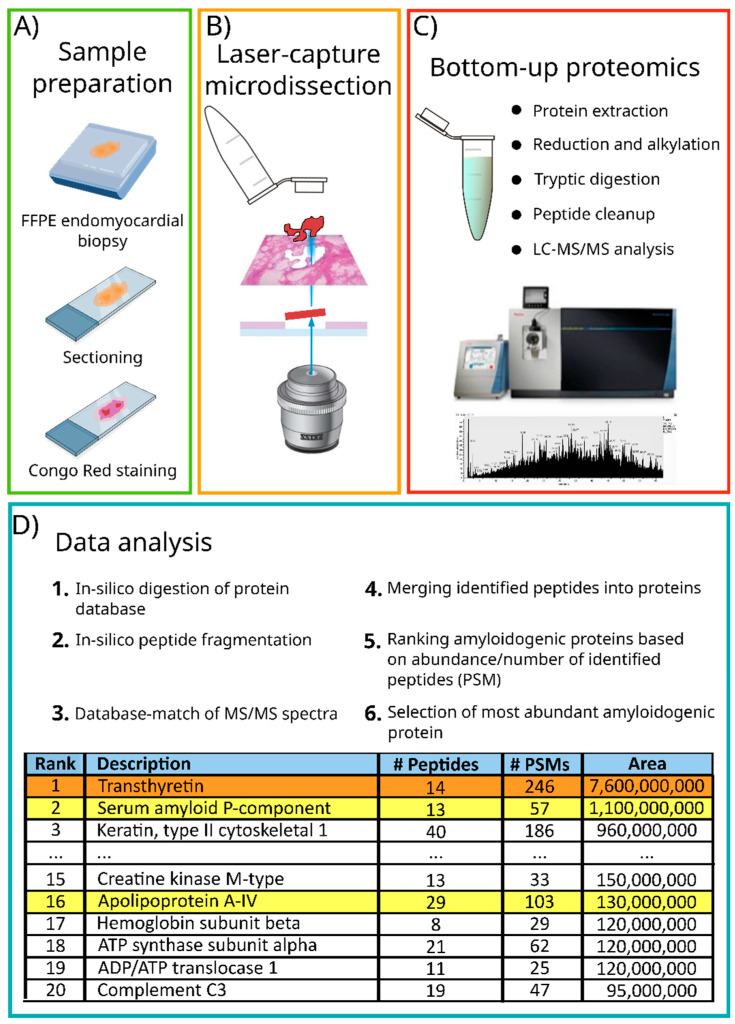
Bottom-up proteomics workflow for amyloid proteins from a formalin-fixed and paraffin-embedded endomyocardial biopsy. (**A**) A histological section is stained with Congo red to identify amyloid deposits; (**B**) Congo red-positive areas are isolated by laser-capture microdissection and processed with a bottom-up proteomics workflow; (**C**) fragmentation spectra are matched with an in silico-generated database; (**D**) the most abundant amyloidogenic protein (evidenced in orange) is identified, while the co-precipitating proteins (in yellow) represent part of a characteristic amyloid protein signature. FFPE, formalin-fixed and paraffin-embedded; LC–MS/MS, liquid chromatography coupled with tandem mass spectrometry.

**Figure 10 biomedicines-10-03054-f010:**
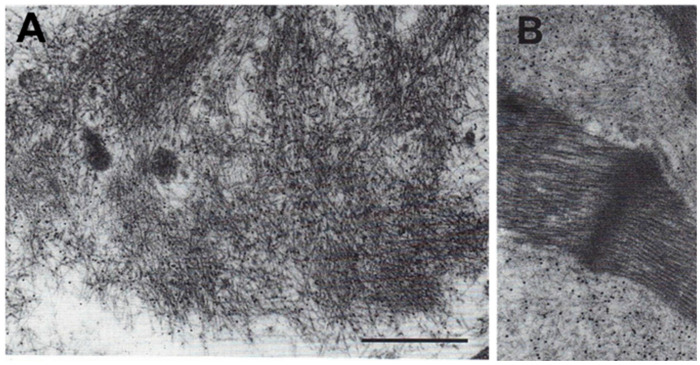
Transmission electron microscopy in cardiac amyloidosis. Amyloid fibrils are shown in a left ventricle endomyocardial biopsy by uranyl acetate and lead citrate staining as non-branched, straight fibrils with 6–13 nm diameter in the myocardial interstitium ((**A**); bar, 100 nm) and adjacent to myocytes (**B**); immunoglobulin light chain Lambda immunoreactivity is demonstrated by post-embedding immunogold technique (**B**), the electron-dense immunogold particles (10 nm diameter) binding to amyloid fibrils [34].

## Data Availability

No new data were created or analyzed in this study. Data sharing is not applicable to this article.

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
