# Peer review of "Tissue Characterization in Cardiac Amyloidosis"

_biomedicines, 2022, doi:10.3390/biomedicines10123054_

Round 1

Reviewer 1 Report

The authors present a review article entitled "Tissue Characterization in Cardiac Amyloidosis" which is very well written.  As a review article I find the manuscript is superficial yet it covers the topic concisely.  My comment is that it is too concise. For example, the review of the abdominal fat pad does not include any history of why it became a site of favor and it's many limitations.  I believe that the authors have missed an opportunity to expand upon the discussion of the fat pad biopsy as many (most) physicians have been taught the fat pad is the "best" site of biopsy.  This reviewer favors the rectal biopsy that is not included in mention of "other sites" in the paragraph preceding 2.1.  The authors do address the rectal biopsy in section 2.3.

Likewise in section 2.1, third sentence: "if correctly performed...."  Please expand your discussion for the reader.  I'd like to know the authors' intent with such a leading statement.  This is common throughout the manuscript: statements made without sufficient follow-up.  As a review article the manuscript is too "simplistic" but it reads very well.

Page 5 begins that an experienced pathologist....techniques .and their possible pitfalls demonstrates my above statement that a statement is made but leaves the reader wondering what pitfalls should they be known!  Again, this is the major problem with the manuscript.  

The same can be found on page 6 in section 5: the first sentence is a general statement with no examples provided.  This is not reasonable for a review article!

Same as above in section 6.  Too many general statements without specific examples.  I do not want to read citation papers 53-55 to find out why fat pad biopsies are less suitable for proteomics..

Figure 9D: table font is too small for easy reading; increase the font size comparative to the font size to the left of the table.

Section 7 on page 10.  This reviewer finds it interesting that TEM first appears and the discussion is again superficial!  TEM was once the gold standard and still is at my institution.  Epon has been discontinued for decades and no longer available for use!  All images in Figure 10 are of very poor quality and must be improved.

Section 9, the sentence following ref [70] reads:  "In the next future...."  What does this mean?  The entire sentence is poorly structured in contrast to most of the manuscript.

My summary:  I enjoyed reading the manuscript and although I have many criticisms, it is much too superficial for a review article and it requires a significant expansion of discussion in all sections.  I do hope the authors can fulfil my requests as the article needs to be published but it is not suitable for publication in its current state.

Author Response

We wish to thank the Reviewer for the very helpful comments and suggestions.

We have revised the article, according to the useful comments, as follows:

  1. Section 2.1. History of the abdominal fat biopsy, including fine-needle and excision biopsy, has been included, quoting also why it became a preferred site such as its limitations.
  2. Gastrointestinal tract (including rectum) biopsies have been mentioned in the paragraph preceding Section 2.1 and in Section 2.3 (Other biopsy sites). We have also expanded the issue regarding gastrointestinal (including rectal) biopsies. We have better explained the impact of the rectal biopsies, in particular for the initial diagnosis of amyloidosis, rectal biopsies being often negative in patients with negative fat biopsy.
  3. Section 2.1 In the third sentence, "if correctly performed...." has been erased and discussion has been expanded. The execution of the abdominal fat biopsy has been better explained, and previous studies on the impact of size, quality and/or processing of fat fragments are reported and discussed. 
  4. Page 5, sentence “The tissue must be examined by an experienced pathologist, confident with the disease, the histological techniques and their possible pitfalls”. The paragraph has been extended, as suggested by the reviewer. The impact of biopsy sample number, processing (e.g. serial sections) and proper stainings such as the experience of the pathologist are cardiovascular pathology are then expanded and discussed.  
  5. Page 6, section 5 (Immunohistochemistry of cardiac amyloidosis). A few examples are provided, regarding the most common pitfalls, i.e., false negative or false positive results, non-adequate antisera/antibodies or and tissue over-fixation. 
  6. Section 6. As for page 6, section 5, also this section has been modified according to the requests. It has been expanded to be adequate to a review article, also as to the limits of fat pad biopsies.
  7. Figure 9D: table font has been increased as requested.
  8. Section 7 on page 10. The TEM section has been revised according to the helpful comments of the reviewer, also regarding the role in diagnosis and typing of amyloid, the currently used resins, and the Figure has been modified and improved.
  9. Section 9, the sentence following ex-ref [70] reading "In the next future...."  has been modified to clarify its meaning.
  10. Expanding the discussion of issues according to the Reviewer requests, has led to an increased number of references that have been obviously updated throughout the text.

Sincerely yours,

Angela Pucci

Reviewer 2 Report

The review entitled “Tissue Characterization in Cardiac Amyloidosis” covers several current approaches to characterize CA. Traditional strategies are mentioned such as tissue biopsy and morphologic aspects. Also, mass spectrometry-based proteomic analysis CA is explored has a potential alternative and complementary methodology which needs further developments and protocols. TEM, immunofluorescence among other methods are also presented. Globally, the review has a good coverage of the current science fields used to study CA tissues.  

Author Response

We wish to thank the reviewer for the very kind consideration of our paper.

We did appreciate these comments.

Sincerely yours,

Angela Pucci

Round 2

Reviewer 1 Report

This reviewer is still not happy with Figure 9, especially "D" as the text is "fuzzy".  This should be corrected as the table is difficult to read.  This is likely due to the resolution of the JPG or the type font that needs to be much sharper.

Author Response

We wish to thank the reviewer #2 for the helpful comments and for the kind consideration of our manuscript.  Accordingly, we have revised the manuscript as follows:

1) Figure 9, particularly "D" text has been modified to improve and ease table reading, also by increasing its resolution;

2) the suggested reference by Hummel K et Al. (Negative Fat Pad Biopsy in Systemic AL: A Case Report Analyzing the Preferred Amyloidosis Screening Test. Diseases 2021) has been added. It is mentioned in section 2.1, pag. 3 and the following sentence has been added “Nevertheless, it has to be pointed out that fat biopsy might miss the diagnosis even in AL cases, a negative result not excluding AL diagnosis [33].” Therefore, the references have been updated in the references list and throughout the text.

Finally, for Figure 1-ATTR+ and For Figure 6-Lambda+ we acknowledged the original source, i.e., the Ref. 15, by Pucci A et Al. JAHA 2021, Open Access) whereas Fig. 7A has been replaced because incorrect (i.e., the same as Fig. 6A).

Sincerely,

Angela Pucci